# Progress in Nanocarriers Codelivery System to Enhance the Anticancer Effect of Photodynamic Therapy

**DOI:** 10.3390/pharmaceutics13111951

**Published:** 2021-11-18

**Authors:** Yu-Ling Yang, Ke Lin, Li Yang

**Affiliations:** State Key Laboratory of Biotherapy and Cancer Center/Collaborative Innovation Center for Biotherapy, West China Hospital, Sichuan University, Chengdu 610041, China; yyuling@stu.scu.edu.cn (Y.-L.Y.); linke@stu.scu.edu.cn (K.L.)

**Keywords:** codelivery nanocarriers, photodynamic therapy, anticancer therapies, combination therapy

## Abstract

Photodynamic therapy (PDT) is a promising anticancer noninvasive method and has great potential for clinical applications. Unfortunately, PDT still has many limitations, such as metastatic tumor at unknown sites, inadequate light delivery and a lack of sufficient oxygen. Recent studies have demonstrated that photodynamic therapy in combination with other therapies can enhance anticancer effects. The development of new nanomaterials provides a platform for the codelivery of two or more therapeutic drugs, which is a promising cancer treatment method. The use of multifunctional nanocarriers for the codelivery of two or more drugs can improve physical and chemical properties, increase tumor site aggregation, and enhance the antitumor effect through synergistic actions, which is worthy of further study. This review focuses on the latest research progress on the synergistic enhancement of PDT by simultaneous multidrug administration using codelivery nanocarriers. We introduce the design of codelivery nanocarriers and discuss the mechanism of PDT combined with other antitumor methods. The combination of PDT and chemotherapy, gene therapy, immunotherapy, photothermal therapy, hyperthermia, radiotherapy, sonodynamic therapy and even multidrug therapy are discussed to provide a comprehensive understanding.

## 1. Introduction of Photodynamic Therapy and Photosensitizers

### 1.1. Photodynamic Therapy

Photodynamic therapy (PDT) is a modern noninvasive antitumor technique that has great clinical application potential due to its advantages of simple operation and low systemic toxicity. PDT has been clinically approved for the treatment of some tumors, such as advanced esophageal cancer and advanced lung cancer [1]. Local or systemic photosensitizers (PSs) can be activated to produce cytotoxic reactive oxygen species (ROS) after absorbing light from an appropriate wavelength laser, which can induce tumor cell necrosis and apoptosis or cause angiotoxicity to block tumor cell nutrient supply. In addition, acute inflammatory responses and immunogenic cell death (ICD) induced by PDT have been shown to activate the body’s immune system and result in the reconstruction of the tumor microenvironment [2,3].

There are two main mechanisms of the photodynamic reaction. When PSs enter cells, light at a wavelength coinciding with the PS absorption spectrum irradiates the issue, and PSs will be converted from the singlet basic energy state into the excited singlet state because of photon absorption. Part of the energy directs a PS molecule to the excited triplet state. In type I reactions, the PSs in their excited triplet state react with biomolecules and ROS can be generated from radical oxygen species formed by electron transfer reactions such as superoxide ions (O_2_^•−^) and hydroxyl radical (OH^•^). In type II reactions, the energy of PSs in their excited triplet state is transferred directly to the oxygen molecule in the basic energetic state, resulting in the generation of singlet oxygen, which is highly reactive and cytotoxic [2,4]. ROS produced by the two reactions irreversibly damage cells and microvessels, and ROS can fight against cancer by inducing apoptotic cell death, a subroutine mode of active immunity [3,5]. Type II is considered as the most important process affecting PDT efficiency. The balance of the contribution of both reactions depends on many factors, including oxygen concentration and on the structural features of the PSs.

### 1.2. Photosensitizers

PDT emerged as a tumor treatment in 1907, and the early progression of it is closely related to the development of PSs [4]. The PS types are generally divided into non-porphyrins and porphyrins. The typical non-porphyrin PSs are hypericin, including phenothiazinium dyes, xanthene dyes, phthalimide derivatives, indocyanine green (ICG) etc. Methylene blue (MB), toluidine blue O, Rose Bengal (RB), ICG, as well as the near-infrared dye IR780-iodide have commonly been used in recent years [6,7,8]. In addition, inorganic materials like graphene oxide (GO) and gold/silver nanoparticles (AuNPs/AgNPs) are also included [9,10,11], the latter have good biocompatibility and tunable optical properties, and ^1^O_2_ can be generated through surface plasmon excitation for PDT of tumors [12]. Metal oxides such as TiO_2_ and ZnO_2_ NPs can be photoactivated to produce ROS through electron-hole pair interactions with the surrounding ^1^O_2_. These nanomaterials generally have the advantages of low toxicity, inertness, good biocompatibility and photostability [13]. Other materials, including black phosphorus nanosheets [14], fullerene [15] and GCNS [16], can also be modified to achieve multifunctionalization.

Porphyrins PSs in the early stage were mainly mixtures obtained from natural porphyrins. For example, hematoporphyrin derivative (HPD), was first used as a PS for bladder cancer in 1970 [17,18]. Unfortunately, Since such PSs had a long half-life and would accumulate excessively in the skin, patients receiving the treatment need to avoid intense light for weeks and their lifestyle is severely affected [19,20]. The second generation are substances with a pure structure, including porphyrins, chlorins, benzoporphyrins and phthalocyanines [9,10]. The common one is 5-aminolevulinic acid (5-ALA), a prodrug of protoporphyrin IX (PpIX), also the current FDA-approved standard for brain tumor visualization. Chlorin e6 (Ce6) is photoactivated at near-infrared wavelengths (664–665 nm) [21,22,23], has a high efficiency of singlet oxygen generation, and can penetrate deep tissue [24]. Visudyne, the active ingredient of which is verteporfin (benzoporphyrin derivative, BPD), is the only FDA-approved NIR PS for intravenous injection with high singlet oxygen yield and a low skin phototoxicity [25]. Verteporfin has poor solubility and rapid clearance speed in vivo [26], so nanocarriers (liposomes) or cosolvents are used to ameliorate these limitations [27]. Phthalocyanines have two main UV-vis absorption wavelengths: 350 nm and 680 nm, whose high intensity is one of the main characteristics of these dyes. Due to the presence of a high Q band at 680 nm, these dyes can be activated by red light with strong tissue penetration [28]. They have low toxicity and rapid elimination properties [29]. The photophysical properties of phthalocyanines are strongly dependent on central metal ions, and complexes with diamagnetic metal ions such as Zn^2+^ and Al^3+^ help to enhance PDT, including zinc phthalocyanine (ZnPc), zinc hexadecafluorophthalocyanine (ZnF_16_Pc), aluminum chloride phthalocyanine (ClAlPc), etc. At present, there have been a variety of phthalocyanine-based PSs in clinical trials, such as CGP55847, Photosens and so on [30]. The third generation of PSs currently under investigation is designed to synthesize substances that have a higher affinity for tumor tissue, in the form of former drugs or those based on novel drug delivery systems (liposome, polymer, or micelle) [17,18,19].

The way PDT induces cell death (apoptosis, necrosis, autophagy cell death, or a combination of them) is related to the localization of PSs in subcellular components. Many reports indicate that mitochondria are important targets of PDT, and PSs targeting mitochondria has been proved to induce apoptosis more effectively, which may be related to the loss of mitochondrial membrane potential, inhibition of ADP/ATP exchange, respiratory enzymes and oxidative phosphorylation [4,20,31]. PSs targeting lysosomes could induce the release of cathepsins upon photodamage, which could cleave pro-apoptotic protein Bid and promote mitochondrial apoptosis, or cleave caspase-3 and inhibit apoptosis [31]. Necrosis is caused by high levels of cell damage and usually requires a higher dose than apoptosis. When PSs is predominantly present in the plasma membrane, this is the main form of cell death. In this type of injury, plasma membrane integrity is lost, cells dissolve, and tissue inflammation is triggered. PDT has been reported to induce autophagy, a mechanism for cell survival or adaptation in which injured cells attempt to repair or remove dysfunctional organelles to promote survival. If the initial response fails, autophagy turns into a cell death signal. PDT-induced autophagy is associated with subcellular sites damaged by ROS and can be triggered by light damage to key organelles such as endoplasmic reticulum, mitochondria and lysosome [32,33].

In addition, the fluorescence capability of PSs is widely used in clinical imaging, especially NIR fluorescence imaging technology, which can depict tumor tissue in real time, thus being used to provide highly sensitive images for surgery or to monitor drug therapy [34].

## 2. Nanotechniques to Improve Photodynamic Therapy

Poor water solubility is a common feature of PSs and limits their clinical application. Chemical coupling combining hydrophobic PS and hydrophilic substances could enhance the circulation time in the blood [24,35]. Physical binding is a more common approach. PSs are loaded into chitosan (CS) [23,36], cationic polymer liposome microcapsules [37,38,39] and other carriers, which can not only improve the physical and chemical properties but also avoid degradation in the physiological environment [40,41,42].

The use of targeted vectors helps reduce toxicity and allows the same therapeutic effect to be achieved with smaller doses of PS. This is mainly related to increasing the selectivity of PSs tissue distribution and improving drug bioavailability [4]. In addition to targeting specific tissues, simultaneous targeting of multiple organelles has also been reported to have a stronger therapeutic effect. This can be achieved by combining PSs with different subcellular localization or by using different nanocarriers [27,43,44].

Hypoxia at the treatment site [45,46], low ROS generation efficiency, and low light penetration are also limitations. Inorganic NPs such as AgNPs can mediate greater ROS production through Ag^+^ ions released from the surface and have been reported to enhance the anticancer effect of PSs such as MB and Ce6 [47,48,49]. GO is also commonly used to enhance ROS production, and studies have found that the use of GO loaded with MB can raise the local tumor temperature to approximately 40 °C, effectively preventing tumor metastasis and regeneration [50]. The use of noncovalently bound graphene oxide dihydrogen porphyrin derivatives as long-wavelength absorbers of PS (λ_max_ of 707 nm) enables light to penetrate deeper into the tumor site, overcoming the limitations of low light penetration [51]. For some hypoxic tumors, oxygen or oxygen generators can be codelivered with the PS to enhance the PDT effect through a self-supply of oxygen. Common oxygen transport carriers include nanoscale artificial red blood cell tumors targeting human serum albumin, which can deliver oxygen-loaded hemoglobin and PSs to the tumor site [52,53]. Perfluorocarbon (PFC), an artificial blood substitute that can dissolve large amounts of oxygen, can also alleviate tumor hypoxia after cotargeted delivery with PS [54]. Another method is to use metal carriers to react with peroxide to generate oxygen at tumor sites [55,56]. For example, an activated system using linoleic acid peroxide (LAHP) and iron oxide NPs (IO NPs) can induce the apoptosis of cancer cells through tumor-specific ^1^O_2_ generation and subsequent ROS-mediated mechanisms [46]. In addition, it is feasible to transform type II oxygen-sensitive photochemical reactions into Type I photochemical reactions [57]. Other approaches to overcome hypoxia include the use of siRNA to inhibit hypoxia-inducible factors.

The upregulation of vascular endothelial growth factor (VEGF) and heat shock protein 70 (HSP-70) are the main causes of PDT tolerance. To obtain a more ideal PDT treatment effect, PS can be delivered together with gene therapy drugs or small molecule inhibitors targeting VEGF or HSP-70 [58]. In addition, NO produced by photostressed tumor cells can induce anti-PDT effect and enhance tumor aggressiveness. This can be addressed by codelivery with iNOS inhibitors. These would be described in detail in the section on codelivery treatment [59,60,61].

## 3. Codelivery of PSs and Anticancer Drugs with Nanoparticles

There are many factors that contribute to decline in photodynamic therapy, including the metastatic tumor at unknown sites, inadequate light delivery, lack of sufficient oxygen and induction of an anti-PDT effect [1]. To overcome these limitations and to improve efficiency, the researchers have combined PDT with other treatments. Codelivery is one of the most common methods to achieve combination therapy and has been widely studied. Currently, common treatment schemes that are amenable to PS codelivery can be divided into the following types: chemotherapy, gene therapy, immunotherapy, photothermal therapy (PTT), hyperthermia (HT), radiotherapy, sonodynamic therapy and multiple drugs codelivery [62,63,64,65,66,67,68,69,70,71] (Figure 1). Here, we will mainly introduce the nanocarriers used for the codelivery of PSs and antitumor drugs and discuss how the drugs enhance each other.

### 3.1. Chemotherapy

Chemotherapy is a commonly used method for tumor treatment, but systemic toxicity and side effects cause patients to face great pain when receiving treatment, and the efficacy is limited by stability, targeting and multidrug resistance [72]. Many reports have proven that the application of targeted delivery vectors can improve the physical and chemical properties of chemotherapy drugs, while combination with PDT can help to improve multidrug resistance and other problems [73,74]. Mechanistically, chemotherapeutic drugs bind to the DNA of tumor cells and block DNA replication, which leads to suppressed cell division and ultimately death [75]. It has also been found that many chemotherapeutic drugs can increase intracellular ROS levels and oxidation-reduction homeostasis of cancer cells, thus enhancing the sensitivity of tumor cells to PDT [76]. On the other hand, light-independent chemotherapy can kill deep-level tumor cells that PDT cannot, while increased tumor vascular permeability induced by PDT can enhance the accumulation of nanomaterials in tumors, thus enhancing the efficacy of chemotherapy [77,78].

Cationic liposomes are widely used in the codelivery system of PSs and chemotherapeutic drugs due to their preferential accumulation in the vascular endothelium [79]. The introduction of porphyrin-phospholipid (PoP), which is a kind of PS-coupled lipid, enables photoprogrammed controlled release of cationic liposomes. ROS produced by irradiation can oxidize unsaturated lipids and accelerates the release of chemotherapeutic drugs [80,81] (Figure 2A). Doxorubicin (DOX) was loaded into PoP liposomes for intravenous injection. Tumor vascular permeability and drug accumulation were significantly increased under near-infrared light, while empty PoP liposomes without drug loading also showed antitumor effects, suggesting that conjugation with phospholipids did not affect the photodynamic effect of PS [82,83]. In the synthesis of PoP liposomes, the selection of PoP and the proportion of other raw materials are crucial to the morphology and serum stability of the liposomes. Generally, a lower content of PoP in the lipid bilayer leads to higher serum stability [84]. Moreover, the dosage needs to be considered. When DOX is loaded in excess, the bilayer of PoP liposomes becomes elliptical due to instability, while loading irinotecan (a camptothecin-derived anticancer drug) does not affect the morphology of the liposomes [62]. In addition, carrier modification can achieve multiple functions; for example, ^64^Cu-labeled POP liposomes can be used for imaging simultaneously with treatment and have been shown to not cause drug leakage [85]. The timing of illumination after POP liposome administration is also important, and studies have shown that the drug accumulates more in tumors at short drug–light intervals [86].

Hypoxia is the cause of drug resistance in both chemotherapy and PDT. The effect of photochemotherapy can be enhanced by codelivery of oxygen carriers [87] or oxygen generators [88] loaded with therapeutic drugs. There are also substances that can serve as both chemotherapeutic agents and oxygen donors. For example, nanoplatinum (Pt), when encapsulated in liposomes with PSs, can provide oxygen as a catalase-like nanoenzyme, while Pt ions leached separately can also exert cytotoxic effects [89]. Although the application of nanocarriers can reduce the toxicity and side effects of chemotherapy drugs, it cannot improve the antitumor tolerance caused by the upregulation of heat shock protein HSP-70 after PDT. Hailong Tian [90] combined quercetin (Qu), a chemotherapy drug with the dual effects of anticancer and heat shock protein inhibition, with IR780. IR780 modified with hydrophilic biotin and Qu were assembled into a delivery carrier in solution, which successfully enhanced the therapeutic effect of PDT by inhibiting the expression of HSP-70. PDT and some chemotherapeutic agents, such as DOX and Pt, can mediate ICD. For example, the combined action of oxaliplatin and PDT was shown to expose calreticulin (CRT) on tumor cells, successfully activating host immunity and creating a suitable microenvironment for subsequent immunotherapy combination [91]. More examples of recently published studies on the codelivery of photosensitizers and chemotherapy drugs based on nanocarriers are illustrated in Table 1.

### 3.2. Gene Therapy

The rapid development of nanodelivery systems has improved the penetration ability of cells and tissues and stability of exogenous genes under physiological conditions, allowing gene therapy technology to break through previous bottlenecks and enable clinical application [102,103]. The combination of gene therapy and PDT offers a high degree of precision based on complementary base pairs, which most other therapeutic combinations lack. Exogenous genes are required to efficiently enter cells and successfully escape intracellular bodies before they can play a therapeutic role, which poses great challenges to the material properties of delivery vectors [104]. Programmable vectors have been proposed to solve this problem to some extent and are commonly released under internal stimuli such as enzymes under special pH conditions. Photochemical internalization (PCI) is an intracellular delivery technique using an exogenous light as a stimulus. After activation of PS, ROS can be rapidly produced within a short time to destroy the membrane of the intracellular body and release therapeutic agents such as nucleic acid drugs in the cytoplasm [105,106] (Figure 2B). This external light-dependent regulation is more controllable and stable than the internal response-dependent system and can achieve higher spatially controlled and targeted gene delivery [107]. Another type of photoprogrammed gene regulation uses PDT active nanomaterials as gene delivery vectors, such as PPBP, the black phosphorus (BP) nanosheets prepared with PEG and PEI modification, a black scale nanomaterial that shows PDT activity under light and then specifically degrades to release siRNA in a high ROS and acidic tumor environment to achieve targeted delivery [108].

Hypoxia influences PDT, while upregulation of VEGF and HSP-70 leads to PDT tolerance. Changing the expression of these proteins by gene therapy can help resolve these challenges to some extent and enhance the sensitivity of tumor cells to PDT. HIF1α is a hypoxia-inducible factor that plays a key role in tumor cell proliferation and angiogenesis [109]. Zheng WH [110] used anisamide-targeted lipid-calcium-phosphate (LCP) nanoparticles to achieve codelivery of protoporphyrin IX (PpIX) and HIF-1α siRNA. The results showed that HIF1α downregulation not only directly inhibited tumor cell generation but also promoted ROS production in the tumor environment, thereby enhancing PDT-mediated apoptosis. Nrf2 is a key antioxidation regulator that prevents ROS accumulation and promotes angiogenesis. Deng S [111] codelivered CRISPR–Cas9 ribonucleoprotein (RNP) with Ce6, ROS generated by the latter caused release of Cas9/sgRNA into cytoplasm, resulting in Nrf2 interference and preventing tumor cells escaping from ROS-mediated killing. In terms of improving PDT-induced tolerance, Jang Y [112] prepared a DOX-siVEGF-NPS/Ce6-MBS complex coloaded with VEGF siRNA and Ce6 for the treatment of squamous cell carcinoma. Under the action of VEGF siRNA, tumor angiogenesis was significantly reduced, and the antitumor effect was improved. Similar results were reported in Cao Y [113], the MnO_2_ nanosheet was first surface decorated with Cu_2−x_S and then loaded with HSP-70 siRNA to form MnO_2_/Cu_2−X_S-HSP-70-siRNA, which mediated the heat shock response and showed superior synergistic antitumor ability. In addition, the use of therapeutic genes to regulate the expression of proteins related to the growth, development, differentiation and metabolism of tumor cells can enhance the effect of PDT or supplement the limitations of PDT. For example, gene therapy can activate the body’s immune system to treat metastatic cancer [114] and inhibit epithelial-mesenchymal transition (EMT) to avoid tumor recurrence [115]. Overall, the diversity of cancer-causing genes and the complexity of pathogenesis, as well as the innate differences of individuals, give gene therapy and PDT to infinite possibilities in combination. More examples of recently published studies on the codelivery of photosensitizers and gene therapy drugs based on nanocarriers are illustrated in Table 2.

### 3.3. Immunotherapy

PDT can activate the body’s immune response in two ways: one is to induce an acute inflammatory response of the host and release various proinflammatory signals; the other is to trigger ICD by injured or dead tumor cells to release damage-associated molecular patterns (DAMPs) and neoantigens as danger signals [3,124]. Although the immune response mediated by PDT is not enough to kill tumor cells, it can create a reconstructed immune microenvironment for further antitumor immunotherapy [63].

Immune adjuvants are a class of immune-stimulating molecules that activate tumor-specific immune responses by interacting with Toll-like receptors (TLRs) on antigen-presenting cells (APCs) [125]. The combination of immune adjuvant and PDT has the dual ability to activate the immune system. A nanometal organic framework (nMOF) formed by direct self-assembly of metal ions and PSs is often used as a codelivery carrier and is characterized by a high loading efficiency and good biocompatibility [126]. Cai Z [127] prepared PCN-ACF-CpG@HA NPs loaded with the immune adjuvant CPG by combining PS tetrakis (4-carboxyphenyl) porphyrin (H_2_TCPP) with zirconium ions to target tumors with high expression of the CD44 receptor. CPG and PDT together mediated a strong antitumor immune response, with significantly higher CD8+ and CD4+ T cell infiltration at the tumor site than that in the control group. Similarly, the cationic W-TBP designed by Ni K [128] based on nMOF can directly adsorb negatively charged CPG through electrostatic action, promote its internalization and DC maturation, and enhance antigen presentation in coordination with PDT-induced CRT exposure. In addition to treating tumors in situ, this codelivery combination kills distant metastatic cancer cells. Xia Y [129] evaluated the efficacy of CPG combined with the PS verteporfin in the treatment of 4T1 metastatic breast cancer; the results showed that the activation of DC cells was significantly increased, and the tumor volume of tumor-bearing mice was smaller than that of other control groups. Xu C [130] also achieved similar results in the treatment of local and metastatic B160F10 melanoma, which was enhanced by promoting DC recruitment and cytotoxic T cell infiltration at the tumor site. Exogenous antigens can also activate immune responses. Ovalbumin (OVA), a commonly used model antigen, can be used as a supplement to immune stimulation induced by PDT. The effect of such codelivery was evaluated in the studies of Huang R [131] and Ding B [132], both of which showed synergistic immune enhancement.

Immune checkpoint therapy has made significant breakthroughs in recent years and can effectively improve the immune system’s response to tumors. At present, several immune checkpoint inhibitors have been approved by the FDA for clinical treatment [133]. Tumor vascular abnormalities and the immunosuppressive tumor microenvironment (TME) caused by indoleamine 2,3-dioxygenase 1 (IDO1) seriously affect the efficacy of PDT-mediated immunotherapy. Codelivery of PSs with IDO1 inhibitors is beneficial for amplifying the effects of photodynamic immunotherapy. Combinations that have been reported include ferritin and polyethylene glycol–PLGA (PEG–PLGA) coloaded with ZnF_16_Pc and IDO inhibitor NLG919 [134] or Ce6, tyrosinase inhibitor axitinib (AXT), IDO1 inhibitor dextro-1-methyltryptophan (1MT) and human serum albumin self-assembled NPs [135]. These combinations can improve the tumor microenvironment by normalizing tumor blood vessels, improving hypoxia levels, promoting the invasion of immune effector CD8+ T cells in tumors, and reducing the immunosuppressive properties of tumors, which represents a promising tumor treatment strategy. Cytotoxic T-lymphocyte-associated antigen 4 (CTLA-4) and programmed death-ligand 1 (PD-L1) are the most common immune checkpoints and are closely related to the immune function of T cells. Xu J [136] used the self-assembly property of Ce6 and immunoglobulin G (IgG) in the nanoscale affinity range to bind Ce6 to aPD-L1 or double bind to αPD-L1 and αCTLA-4 in the immunocheckpoint blocking treatment of glioma in situ and colon cancer. This combination therapy successfully prolonged the survival of tumor-bearing mice and produced a long-term memory response, avoiding tumor recurrence. In addition, zinc phthalocyanine and aCTLA4 have been coadded into microneedles prepared by hyaluronic acid and dextran for skin cancer delivery. This minimally invasive percutaneous drug delivery platform can also effectively induce an antitumor immune response and avoid the systemic distribution of drugs to reduce toxicity and side effects [137]. Without affecting coloaded drugs, nanodelivery systems targeting important organelles such as mitochondria [138] and the endoplasmic reticulum [139] can be designed to enhance PDT-triggered ICD, thereby enhancing immune activation (Figure 3).

Photoimmunotherapy is a tumor-targeting therapy using specific antibodies to tumor-associated receptors chemically coupled with PSs, which is more accurate than conventional PDT and is suitable for tumors at sensitive anatomical sites. Hasan T [140] coupled epidermal growth factor receptor (EGFR) monoclonal antibody Cetuximab with benzoporphyrin derivative for pancreatic ductal adenocarcinoma treatment. The results showed that the photoimmune nanoconjugate (PIN) had a high binding specificity and could rapidly penetrate heteromorphic organoids, providing approximately 16-fold enhancement in molecular targeted NIR photodestruction. Nevertheless, single-receptor targeted therapy may cause tumor subsets with low receptor expression to evade treatment and thus fail to completely ablate tumors. Hasan T [141] further constructs a triple receptor-targeted PIN (TR-PIN), cetuximab, holo-transferrin, and trastuzumab conferred specificity for EGFR, transferrin receptor (TfR), and human epidermal growth factor receptor 2 (HER-2). Researchers compared the binding ability of TR-PIN to tumor cells with different levels of receptor expression (EGFR, TfR or HER-2), and found that TR-PIN has the ability to recognize multiple tumor targets, effectively photodynamically eradicating different tumor subsets and reducing escape. More examples of recently published studies on the codelivery of photosensitizers and immunotherapy drugs based on nanocarriers are illustrated in Table 3.

### 3.4. Photothermal Therapy (PTT)

Additionally, as a minimally invasive treatment, PTT has certain similarities with PDT. Photothermal agents concentrated at the tumor site absorb laser radiation energy and convert light energy into hyperthermia, resulting in the thermal ablation of adjacent cells. PTT enhances the effect of PDT, mainly by improving blood flow and increasing oxygen content in tumors [146] (Figure 4A).

Another more conventional codelivery method is to simultaneously load the photothermal agent and PS or prodrug of PS in nanodelivery systems such as micelles, vesicles, and liposomes. Gang Chen et al. [151] developed CS NPs as codelivery carriers of photothermal agents (IR780) and prodrug of protoporphyrin IX (5-aminolevulinic acid, 5-ALA) for oral administration in the treatment of subcutaneous colon cancer in mice. CS keeps the drug stable even under acidic conditions in the stomach, allowing the drug to successfully accumulate at the tumor site. Mechanistic studies have shown that the oxidative stress response at the tumor site is enhanced, producing more ROS, superoxide and ^1^O_2_. The enhanced effect of this synergistic administration of light and heat on cancer treatment was also reported in the research of Xiaodong Liu [65]. The high singlet oxygen generation capacity and photothermal conversion efficiency make this treatment strategy more severely phototoxic to both superficial and deep tumor cells.

When the excitation wavelengths of the PS and photothermal agent were different, the complexity of the treatment was greatly increased. To solve this problem, Jing Lin [152] designed a gold vesicle with a strong plasma coupling effect. This gold vesicle was densely packed with monolayer gold NPs, and Ce6 was encapsulated inside. The enhancement of plasma coupling between gold NPs in close proximity lead to a redshift in extinction spectra. Therefore, the photothermal and photodynamic effects could be stimulated simultaneously by a single 671 nm laser irradiation. Some special materials, such as nano GO, themselves have the dual nature of promoting both PDT and PTT, and their development for PTT can reduce the complexity of material preparation [153]. More examples of recently published studies on the codelivery of photosensitizers and photothermal agents based on nanocarriers are illustrated in Table 4.

### 3.5. Hyperthermia Therapy (HT) and Magnetic Hyperthermia Therapy (MH)

As one of the emerging noninvasive treatment options, HT may enhance PDT by alleviating hypoxia [168,169]. The mechanism is that as temperature increases, blood flow at the tumor site increases and microcirculation improves, thereby increasing tumor oxygenation [170]. MH, also known as magnetic therapy, is based on the heat generated by magnetic NPs (MNPs) under the action of an alternating magnetic field (AMF) to target and kill tumors without harming surrounding healthy tissue [171]. The emergence of MH has made the combination of HT and PDT a possibility, and the generated heat affects deeper tumors, which is beneficial for enhancing PDT [172] (Figure 4B). In terms of codelivery, magnetic nanomaterials can be directly used as carriers or loaded with PS in other carriers. Hongwei Gu [66] conjugated magnetite Fe_3_O_4_ with porphyrin derivatives, which is highly efficient and has a low systemic toxicity. Huang WC [173] used monocytes derived from bone marrow as carriers to jointly transport Ce6 and oxygen superparamagnetic iron oxide NPs for combined therapy. This is an interesting combination strategy, but the relevant literature is still limited, and researchers need to conduct more in-depth research and discussions. More examples of recently published studies on the codelivery of photosensitizers and hyperthermia agents based on nanocarriers are illustrated in Table 5.

### 3.6. Radiotherapy

Band gap materials convert X-rays into light photons in the UV-vis region, which can be used to activate PSs. This is the basis of a combination of PDT and radiation therapy known as X-ray PDT (XPDT) [177,178]. The use of light is one of the limitations of PDT, and the strong tissue penetration of X-rays can compensate for this shortcoming. The method of operation delivers scintillators and PSs together to the tumor site. Under the action of X-rays, scintillators emit persistent light to activate PSs. This strategy can also produce good therapeutic effects for deep tumors [179], and the therapeutic effect depends to some extent on the energy transfer efficiency of the scintillator. The primary choices of scintillators are rare earth materials [180,181,182] with high photon conversion rates and some metallic materials [183], in addition to a few nonmetallic materials [184] and even quantum dots [185].

Another factor affecting the therapeutic effect is the distance between the PS and the scintillator. One approach is to physically package PS in a coating bound to the scintillator by electrostatic or hydrophobic means or to load both the scintillator and PS into NPs [186,187]; for example, the combination of protoporphyrin IX (PpIX) and scintillator LaF_3_:Ce^3+^ coloaded in PEG-PLGA was shown to exhibit highly efficient loading and high stability under physiological conditions [188]. Physical combination has a certain risk of missing the target, whereas chemical bonding based on covalent bonds or conjugation is complex but more stable. K. K. Popovich [67] coated CeF_3_:Tb^3+^ with PpIX-coupled SiO_2_. The monodisperse SiO_2_ coating had high energy transfer efficiency and protected the scintillator encapsulated within it [189].

The combination of a scintillator and PS solves the problem of light sources, but it is still subject to the strong oxygen dependence of PDT. Zhang C [190] was inspired to integrate scintillators and semiconductor quantum dots with unique optical properties into ionizing radiation-induced PDT synchronous radiotherapy. The addition of a PS without traditional effects reduces the oxygen dependence of the complex, which can subsequently produce a process similar to type I PDT under ionizing radiation. Chuang YC [191] introduced the annealing process to achieve PS-free PDT. The yttrium oxide nanoscintillation complex coated with a silica shell (Y_2_O_3_:Eu@SiO_2_) was subjected to X-ray irradiation and subsequent annealing, and then, photodynamic effects were promoted to mediate tumor cell damage in conjunction with radiotherapy.

Radionuclide-mediated Cerenkov luminescence is a phenomenon produced by the interaction of high-speed charged particles with the surrounding medium. The codelivery of radionuclides with PSs has been reported in several studies to have antitumor effects and prolong the survival of tumor-bearing mice. In addition to mediating Cerenkov luminescence, radionuclides can directly kill tumor cells [192,193]. Although this combination has achieved some success in the treatment of deep tumors, there are still many shortcomings to be overcome, such as the blueshift of the low-energy radiation emission spectrum to the ultraviolet region. More examples of recently published studies on the codelivery of photosensitizers and scintillators/radionuclides based on nanocarriers are illustrated in Table 6.

### 3.7. Sonodynamic Therapy (SDT)

Sonodynamic therapy (SDT) uses ultrasound and acoustic sensitizers to treat cancer. Ultrasound, on the one hand, activates ultrasonic-sensitive species gathered in tumor cells through transient sonoluminescence to produce ROS. Hot spots from ultrasonic radiation, on the other hand, release energy high enough to cause bubbles to form and oscillate, resulting in permanent cell damage. The latter phenomenon is also defined as sonography-induced cavitation effects that together lead to tumor cell apoptosis or necrosis [200,201,202,203]. The extent of tumor cell death depends on the intensity and frequency of ultrasound and the duration of exposure, so it is controllable, safe and noninvasive, similar to PDT.

The strong penetration ability and low tissue attenuation of ultrasound make sonophotodynamic therapy (SPDT) have a wider range of indications. PDT is suitable for superficial cancer types such as skin cancer [204] and esophageal cancer [205], but the treatment effect for melanoma skin cancer has not been ideal, which is caused by the absorption of visible light by melanin to remove ROS [206]. In contrast, SDT was significantly more effective for melanoma than PDT, although there was no differences between the two treatments for common skin cancer [207].

The ultrasonic activation ability of PSs has also been verified to some extent. Some PSs have both SPDT activity, which are referred to as photoacoustic agents. We only need to use a specific wavelength of light and a specific frequency of sound to activate the photoacoustic agent so that the photoacoustic combination can be achieved without considering the coordination of physical and chemical properties in the process of multidrug combination. Currently, porphyrins as well as Ce6, Photolon™/Fotolon™, and Sonoflora 1 Sonnelux-1 have been verified [208,209,210]; several studies have reported their therapeutic efficacy and safety, and there are many potential PSs whose ultrasonic activation ability is being verified, which will also find applications in photoacoustic combination in the near future [211].

The modification of codelivery vectors helps to achieve multifunctional and integrated photoacoustic therapy with high spatial activation and preferential treatment of deep penetration. Hong L et al. [68] prepared Ce6 and the high oxygen carrying capacity carrier perfluoropolyether into a nanoemulsion and tested the ROS generation ability at different tissue depths after the application of ultrasound or light. The results showed that PDT had a high degree of spatial selectivity for surface and endoscopically accessible areas, which made it more suitable for the treatment of tumors in vital organs, such as brain cancer. SDT, on the other hand, had the same ROS production efficiency for tissues at different depths. SPDT was conducive to targeting different depths and meeting specific spatial accuracy requirements. Accordingly, sensitive drugs can be combined with tissue-engineered scaffolds to kill tumor cells after surgery. The results showed that SPDT had fewer toxic side effects than conventional chemotherapy, so it has great potential [212]. More examples of recently published studies on the codelivery of photosensitizers and sonophotosensitizers based on nanocarriers are illustrated in Table 7.

### 3.8. Multidrug Codelivery

Cumbersome drug delivery processes, limited efficiency, slow release and low efficacy are common characteristics of multidrug codelivery. Therefore, the main research direction is still focused on solving the problem of coloading and controlled release.

Chen Y [70] designed an NIR-sensitive nanocomposite DLA-UCNPs@SiO_2_-C/HA@mSiO_2_-DOX@NB. This complex breaks the chemical bond and releases paclitaxel (PTX) when irradiated by NIR light at 980 nm, which is then activated by visible light at 450–480 nm to exert synergistic photodynamic and photothermal therapeutic effects. Similarly, the Fe_3_O_4_/g-C_3_N_4_@Ppy-DOX nanocomposite prepared by Cheng HL [4] and graphite-like carbon oxide (G-C_3_N_4_) can not only generate O_2_ through photocatalytic degradation to improve the hypoxic state of solid tumors but also enable the loading of PTX into its mesoporous structure. Polypyrrole, as a photothermal agent, was shown to enhance the antitumor effect of chemo-PDT.

CuS NPs with both PTT and PDT activities were used to achieve the triple combination of PTT-PDT-HT [216]. Curcio A [217] prepared a maghemite (γ-Fe_2_O_3_) nanoflower-like multicore nanoparticle conceived for MH and coated it with a spiky copper sulfide shell (IONF@CuS) for PTT and PDT. This combination showed a good therapeutic effect and reduced the dose of maghemite, which meant lower toxicity. Simultaneous activation of PDT and PTT by X-ray radiation-induced scintillator luminescence can be triply magnified when combined with radiotherapy. Luo L [218] conjugated a scintillator complex and a gold nanorod nanosensitizer. Lanthanide complexes can provide excellent luminescence under X-ray excitation, while gold nanorods can be used not only as PSs for PTT but also as radiosensitizers for enhanced radiotherapy due to their strong near-infrared light and X-ray absorption capacity. As expected, the group treated with both laser and X-ray irradiation showed the best synergistic effect, with significantly more effective tumor ablation than that in the other monotherapy groups.

Low-dose docetaxel has been reported to mediate ICD-activated immune effects [219], and this function of PDT is also mentioned above. Therefore, folic acid-modified mesoporous CuS NPs coloaded with DOX, polyethylenimide-PpIX (PEI-PpIX) and CPG for the treatment of cold tumor breast cancer can effectively rebuild the tumor microenvironment and promote the invasion of cytotoxic T lymphocytes (CTLs) [220]. In addition, it was reported that a MOF was used to integrate chemotherapy-phototherapy-photothermo-immunotherapy. Cus PpIX and DOX were loaded into the core and shell of the metal-organic skeleton ZIF-8, respectively, while CPG was adsorbed outside the shell. This design effectively realized the organic unification of antitumor and antirecurrence/metastasis [221]. More examples of recently published studies on the codelivery of multidrugs based on nanocarriers are illustrated in Table 8.

## 4. Outlook and Discussion

Complex microenvironments, abnormal growth rates and drug resistance during treatment make cancer treatment difficult. PDT, as a new treatment scheme, has the advantages of simple operation, noninvasiveness, safety and few side effects. The emergence of nano-carriers has made great contributions to the development of photodynamic therapy, and the limitation of physical and chemical properties of PS in PDT was solved. In addition to therapeutic effects, PS-mediated fluorescence can also be used for optical imaging to detect tumors, or as image guidance for surgery [230]. The major issues contributing to the failure of PDT are metastatic tumor at unknown sites, inadequate light delivery and lack of sufficient oxygen. This limitation can be addressed by multidrug combination. The combination of PDT and other treatment schemes, such as chemotherapy and radiotherapy, immunotherapy, gene therapy and PTT, has obvious advantages over the single application of one treatment scheme. With the development of nanodelivery vectors, codelivery has become a common method of multidrug combination. In this paper, we review the research progress of PDT combined with other therapies to improve the therapeutic effect on tumors based on the codelivery of nanodelivery carriers.

The combination of PDT and chemotherapy can significantly reduce the dose of chemotherapy drugs, overcome the deep-seated tumors that PDT cannot kill by multidrug resistant chemotherapy, and increase the level of ROS in the tumor microenvironment to enhance the sensitivity of cancer cells to PDT. In combination with immunotherapy, the PDT-mediated inflammatory response and ICD activate the body’s immune response, which can be further enhanced by codelivering an immune adjuvant CPG or tumor antigen, providing a reconstructed immune microenvironment for tumor immunotherapy. PDT can be codelivered with therapeutic genes that target the regulation of hypoxia, angiogenesis, and heat shock proteins, alleviating the tolerance of long-term PDT therapy. Combined with therapeutic genes that target the regulation of other key proteins in tumor growth and metabolism, multimechanism antitumor effects can be achieved. The combination of PDT and PTT can synergistically increase the oxidative stress response at the tumor site, and the PTT-mediated increase in reactive oxygen species levels at the tumor site enhances the sensitivity of tumor cells to PDT. Studies on the combination of PDT and HT are still scarce. Although the discovery of MH is of great help to this codelivery strategy, more studies are still needed to confirm any synergistic effects. PDT combined with radiotherapy can overcome the limitation of PDT in light and can significantly improve the therapeutic effect of PDT on deep tumors. Combined sonodynamic therapy is beneficial to achieve accurate treatment, while satisfying different depths and is suitable for tumor treatment of fine organs. Multidrug codelivery can realize the organic unity of antitumor and antimetastatic effects, as well as recurrence through different antitumor mechanisms.

Although these different antitumor combinations have shown good results, they have their own limitations. PDT has been reported many times to induce sustained systemic immunosuppression, but the current mechanism has not been clarified. The speculated reasons are as follows: (1) ROS produced during PDT inactivate DAMPs released by apoptotic tumor cells, thus failing to stimulate immunity; and (2) apoptotic tumor cells release IL-10, TGF-beta and other immunosuppressive cytokines that affect the generation of CD8+ T cells [231]. This immunosuppression will affect not only the subsequent treatment effect of PDT but also the combination of PDT and immunotherapy. The combination of PDT and PTT is dependent on laser irradiation, which has a poor effect on deep tumors. The combination of PDT and radiotherapy is also subject to the strong oxygen dependence of PDT. Multidrug combination delivery is limited by the development of delivery vectors, drug loading and release. These problems are one of the reasons why codelivery systems based on nanocarriers have not been applied in clinical applications, while other reasons include incomplete safety assessment of the preparation protocol and difficulty in large-scale clinical applications of the preparation.

In summary, this paper focuses on the synergistic enhancement of antitumor therapeutic effects by codelivery combinations of PDT and different treatment regimens. This type of approach is definitely a potential therapeutic strategy, and the existing problems mentioned above need more research to resolve.

## Figures and Tables

**Figure 1 pharmaceutics-13-01951-f001:**
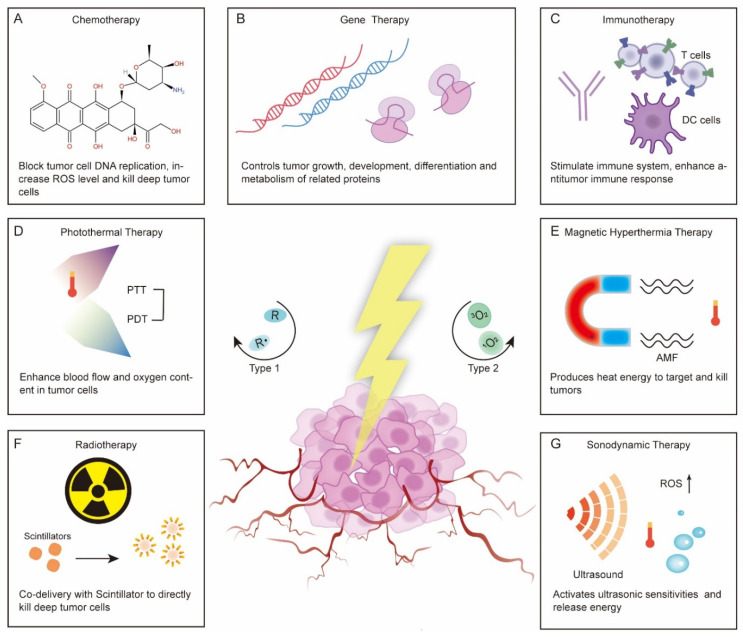
Combination of photodynamic therapy with various therapeutic regimens and possible synergistic antitumor mechanisms. (**A**) Chemotherapy, (**B**) Gene therapy, (**C**) Immunotherapy, (**D**) Photothermal therapy, (**E**) Hyperthemia therapy/Magnetic hyperthermia therapy, (**F**) Radiotherapy, and (**G**) Sonodynamic therapy.

**Figure 2 pharmaceutics-13-01951-f002:**
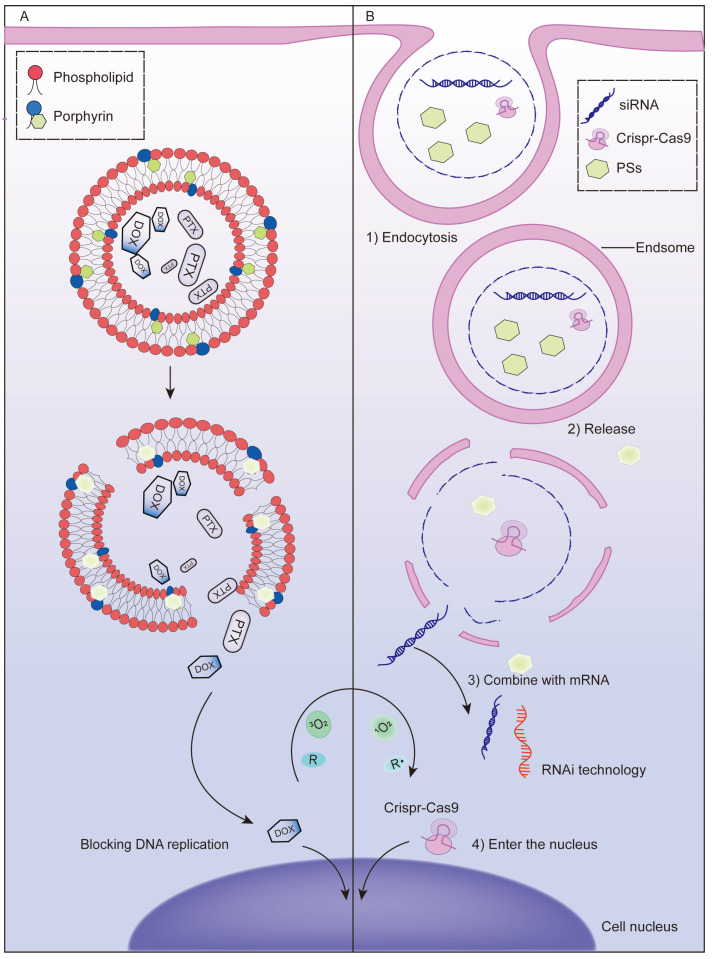
Mechanism of chemotherapy or gene therapy synergistic with PDT to enhance the antitumor effect. (**A**) Diagrammatic illustration of the mechanism of PDT-mediated controlled release of chemotherapy drugs. ROS produced by irradiation can oxidize unsaturated lipids and accelerate the release of chemotherapeutic drugs from porphyrin-phospholipid liposomes. (**B**) Diagrammatic illustration of the mechanism of photochemical internalization (PCI)-mediated endosome escape of nucleic acid drugs. When a photosensitizer is activated by an exogenous light, ROS can be produced rapidly in a short time, destroy the membrane of the intracellular body and release therapeutic genes in the cytoplasm.

**Figure 3 pharmaceutics-13-01951-f003:**
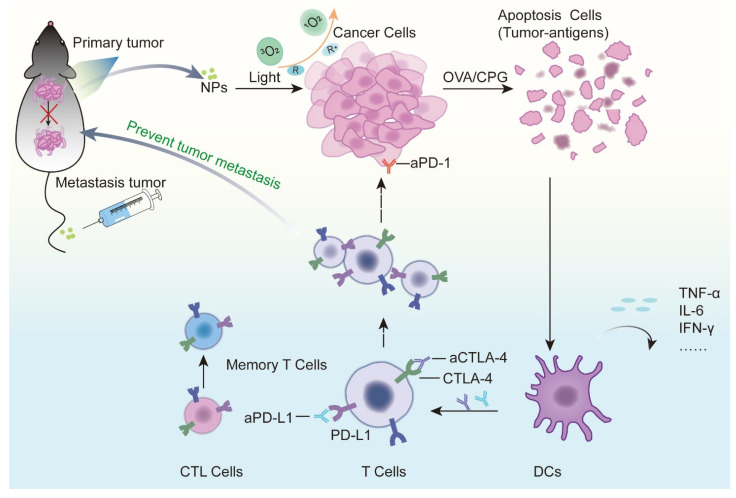
Mechanism by which photodynamic therapy and immunotherapy synergistically enhance antitumor effects. When photosensitizers in tumor sites are activated, they can cause acute inflammation and induce cell apoptosis or necrosis. Dendritic cells mature when stimulated by cytokines released at the site of inflammation and provide antigens to T lymphocytes in regional lymph nodes. Activated T lymphocytes become effector T cells, which are attracted to chemokines, migrate to the tumor and kill tumor cells. Different types of immunotherapy drugs play a role in different steps of the complete antitumor immune cycle. Codelivery of a photosensitizer with an immune adjuvant or tumor antigen can synergistically enhance the activation of the host immune system and improve the immunosuppressive microenvironment. The codelivery of a photosensitizer with CTLA-4 and PD-L1 monoclonal antibodies can enhance the antitumor immunity effect of T cells. The activation of these immune cells also plays a role in preventing metastasis and recurrence.

**Figure 4 pharmaceutics-13-01951-f004:**
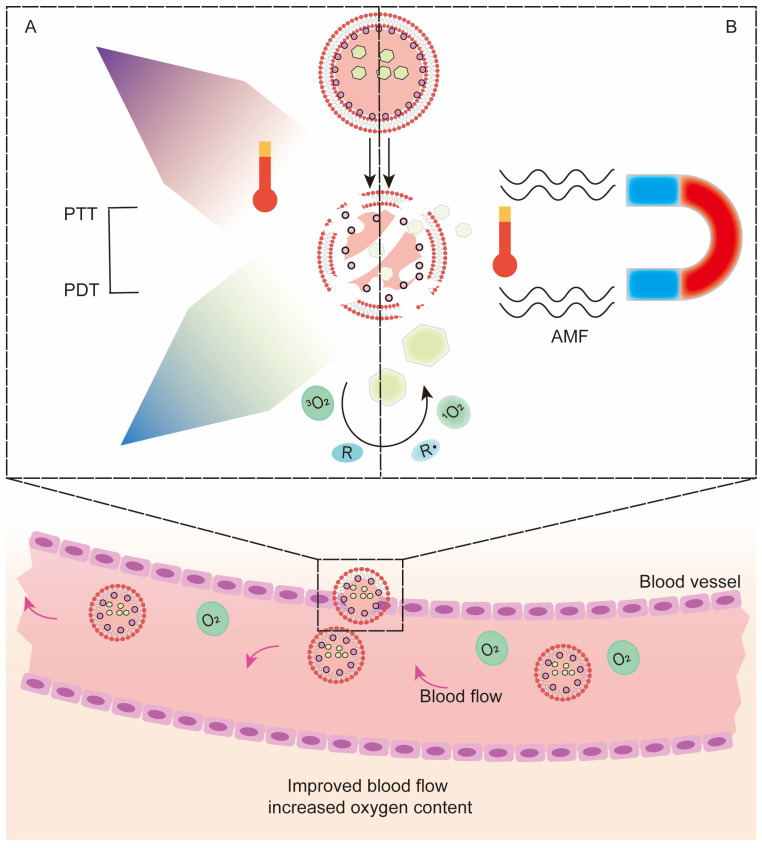
Mechanism of photodynamic therapy combined with photothermal therapy, hyperthermia and magnetic hyperthermia in synergistic enhancement of the antitumor effect based on codelivery systems. (**A**) When light-activated photothermal agents, (**B**) or alternating magnetic fields are applied to magnetic nanoparticles, heat is generated at the tumor site, increasing blood flow and synergistically enhancing PDTSome photothermal agents of inorganic materials themselves can be used as nanocarriers, which makes the combination of PTT and PDT convenient. Gold nanorods have been used as carriers with photothermal conversion capability, and the joint delivery of DOX and PS ICG can simultaneously achieve chemo-PTT-PDT triple therapy [69]. Gold nanocages [147] show strong absorption in the NIR region, and their empty interior and porous walls are suitable for encapsulating PSs. Bo Tian et al. [148] coupled Ce6 on PEG-functionalized GO (GO-PEG-CE6). The photothermal effect of GO promoted the transfer of Ce6, and its destruction effect on cancer cells was significantly better than that of free Ce6. The photothermal agent plasma copper sulfide (Cu_2_-XS) has also been reported to have photodynamic properties [149], allowing the combination of PTT and PDT to be achieved simultaneously, and has been verified in in vitro cultured melanoma cells and mouse melanoma models. The poor photostability and potential long-term toxicity of inorganic nanomaterials limit their clinical application, so more materials with good biocompatibility and high stability have been developed as alternatives. The conductive polymer polypyrrole is a material with relatively high biocompatible photostability and photothermal conversion efficiency, using polyacrylamide (PAH), polyacrylic acid (PAA) and AlPCS_4_. Further modification of polypyrrole resulted in a more stable AlPCS_4_@PPyCONH-PAH-PAA nanoneedle complex in a physiological environment. The results showed an enhanced synergistic effect, with tumor ablation in mice 14 days after treatment and no recurrence within 30 days [150].

**Table 1 pharmaceutics-13-01951-t001:** Codelivery of photosensitizers and chemotherapy drugs based on nanocarriers.

Nanoparticle	Photosensitizers	Chemotherapy Drugs	Tumor	Data Sources	Findings	Ref
Azobenzene-containing conjugated polymers-camptothecin-chlorin e6 NPs (CPs-CPT-Ce6)	Chlorin e6 (Ce6)	Camptothecin (CPT)	HeLa	In vitro, Animals	The –N=N– functional groups in azobenzene could be reduced and cleaved by AZO reductase in tumor hypoxia, promoting ROS production. Controlled release of CPT through the reduction of CPs by AZO reductase.	[92] 2018
Cyclic pentapeptide cRGDfk and Chlorin e6 conjugated silk fibroin (SF)-based NPs	Chlorin e6 (Ce6)	5-Fluorouracil (5-FU)	MGC-803	In vitro, Animals	cRGDfk specifically targets and binds overexpressed α_v_β_3_ integrin receptors on MGC-803 cells to increase tumor aggregation of the drug. Induced high reactive oxygen species generation and produced a good antitumor effect.	[93] 2018
DOX- and perfluorocarbon (PFC)- loaded fluorinated aza-boron-dipyrromethene (PDNBF) NPs	Fluorinated aza-boron-dipyrromethene (NBF)	DOX	4T1	In vitro, Animals	High doxorubicin loading efficiency (25%). PDNBF NPs could be effectively enriched at the tumor site, and DOX could be explosively released by laser irradiation. Significantly inhibited tumor growth and mediated in vivo ultrasound and photoacoustic imaging.	[94] 2020
Pluronic F127 encapsulated halogenated boron-dipyrromethene NPs (LBBr_2_ NPs and LBCl_2_ NPs)	Halogenated boron-dipyrromethene (BDPBr_2_ and BDPCl_2_)	Lenvatinib	Hep3B, Huh7	In vitro	Improved the water solubility of drugs Controlled release through pH response and enhanced the targeting of chemotherapy drugs. Significantly inhibited tumor growth by chemotherapy/photodynamic cotherapy.	[95] 2021
Lactobionic acid-catalase-cis-aconitic anhydride-linked doxorubicin @ chlorin e6 (LA-CAT-CAD@Ce6)	Chlorin e6 (Ce6)	cis-Aconitic anhydride-linked doxorubicin (DOX precursor)	EMT6	In vitro, Animals	Lactobionic acid acted as an active targeting ligand to increase cellular internalization. Controlled release through pH response. Decreased the expression of hypoxia-inducible factor-1α and improved the therapeutic effect.	[96] 2020
Pyropheophorbide a–polyethylene glycol 2000 (Ppa-PEG2k)	Pyropheophorbide a (Ppa)	ROS-responsive oleate prodrug of paclitaxel (PTX)	A549, 4T1	In vitro, Animals	Enhanced the loading efficiency and avoided the quenching effect (ACQ) of PSs. ROS produced by PDT controlled the release of chemotherapy drug PTX.	[97] 2019
Poly (oligo (ethylene glycol) methacrylate)-Paclitaxel @Chlorin e6 NPs	Chlorin e6 (Ce6)	B-sensitive polymer-paclitaxel (PTX)	T24	In vitro, Animals	Photointernalization (PCI) accelerated the uptake of NPs by tumor cells. Tumor growth was significantly inhibited in a PDX model, and the inhibition rate was more than 98%.	[98] 2021
Polyethylene glycol-peptide-poly(ω-pentadecalactone-co-N-methyldiethyleneamine-co-3,3′-thiodipropionate) (PEG-M-PPMT) nanoparticles (NPs)	Chlorin e6 (Ce6)	Sorafenib (SRF)	A549	In vitro, Animals	Increased serum stability of Ce6 and SRF and enhanced drug aggregation in tumors by EPR. Overexpressed MMP-2 in tumor extracellular matrix could partially shed PEG from NPs and form smaller particles that penetrate into tumor tissue. Acidic pH and high intracellular ROS levels accelerated drug release and rapidly killed tumor cells.	[99] 2020
(Phenylboronic acid_4_-E_2_E)_2_-Protoporphyrin IX-(Lipoic acid)_2_	Protoporphyrin IX (PpIX)	Paclitaxel (PTX)	A549	In vitro, Animals	PTX blocks mitosis and makes cells stay at G2/M, prolonging the destruction time of nuclear membrane and promoting the accumulation of photosensitizer in the nucleus. PDT enhanced the internalization and release rate of PTX by destroying lysosomes.	[100] 2020
Poly(ethylene glycol)-b-PMPMC-g-paclitaxel-g-PyTPE micelles (PMPT)	PyTPE, TB	Paclitaxel-SS-N_3_ (PTX-SS-N_3_)	HeLa	In vitro, Animals	Increased uptake of drugs by cells through photochemical internalization (PCI). The high expression of intracellular glutathione was used to break disulfide bonds and induce the release and aggregation of PTX to induce the change in aggregation state of luminescence.	[101] 2021

**Table 2 pharmaceutics-13-01951-t002:** Codelivery of photosensitizers and gene therapy drugs based on nanocarriers.

Nanoparticle	Photosensitizers	Gene Therapy Drugs	Tumor	Data Sources	Findings	Ref
Chlorin e6-DNAzyme/Cu(I)1,2,4-tri-azolate nanoscale coordination polymers (Ce6-DNAzyme/[Cu(tz)] CPs)	Chlorin e6 (Ce6)	Early Growth Response Factor-1 (EGR-1) targeted DNAzyme	MCF-7	In vitro, Animals	[Cu(TZ)] CPs and Ce6 induced type 1 and type 2 PDT at 808 nm and 660 nm, respectively, enhancing the therapeutic effect in hypoxia environment (tumor regression rate was 88.0%). Ce6-DNAzyme was degraded by glutathione overexpressed in tumors, resulting in DNAzyme targeted release and catalytic cleavage of EGR-1 mRNA.	[116] 2021
Cationic guanidylated porphyrin/siRNA complexes (H2-PG/siRNA)	Porphyrin	Inhibitor of apoptosis siRNA (siIAP)	MDA-MB-231	In vitro	The H2-PG/siRNA complex promoted siRNA internalization. Cationic guanidylated modification did not affect the PDT activity of porphyrin.	[117] 2019
CaCO_3_ layer modified MnO_2_ NPs (Mn@CaCO_3_)	Indocyanine green (ICG)	Programmed death ligand 1 siRNA (siPD-L1)	Lewis	In vitro, Animals	Increased oxygen production mediated by MnO_2_ and reduced H^+^ consumption mediated by CaCO_3_ jointly enhanced the PDT effect. siPD-L1 silencing and PDT together activated the immune system, mediating a powerful antitumor immune response.	[118] 2019
(PGL-NH2)2 and fluorocarbon inert gas of C3F8 grafted cationic porphyrin microbubbles (CpMBs)/siRNA	Porphyrin	Pioneer transcription factor 1 siRNA (siFOXA1)	MCF-7	In vitro, Animals	Enhanced drug loading of photosensitizer and siRNA. Under the guidance of contrast-enhanced ultrasound, the accumulation of local microbubbles in tumors was significantly increased through the ultrasound-induced superpore effect. The codelivery system showed superior breast cancer inhibition.	[119] 2018
(NaYF_4_:Yb, Er) upconversion NPs (UCNPs)	5,10,15,20-tetrakis (1-methyl pyridinium-4-yl) porphyrin (TMPyP_4_)	ssDNA with chitosan aptamer (AS1411) and chitosan-targeted DNAzyme	MCF-7	In vitro, Animals	The multivalence of the ssDNA endowed the UCNPs with high recognition and loading capacity of TMPyP4 and DNAzymes.	[120] 2020
Versatile function of cationic phosphonium-conjugated polythiophenes	Polythiophenes	Luciferase gene siRNA	MDA-MB-231	In vitro	The vector effectively self-assembled with siRNA and mediated effective gene silencing (35% and 52% gene silencing efficiencies, respectively). After successful delivery of siRNA, the photoactivity of the vector was restored, which could mediate further PDT.	[121] 2020
Cationic polyporphyrin vectors	Porphyrin	Hypoxia-inducible factor-1α siRNA (siHIF-1α)	H22	In vitro, Animals	Cationic photosensitive drugs were polymerized by an ROS splitting linker to achieve separation in space and avoid ACQ effects and could be used as a carrier to deliver siRNA. Mediated effective intracellular internalization and gene silencing efficiency (60%), synergistic enhancement of phototoxicity.	[122] 2021
Periodic mesoporous ionosilica NPs (PMINPs)	Tetrasilylated porphyrin precursor	Luciferase gene siRNA	MDA-MB-231-Luc-RFP	In vitro	PDT effectively induced cell death with 95% mortality. As a delivery carrier of siRNA, it can successfully escape from endosomes and play a therapeutic role, with gene silencing efficiency up to 83%. It presented triple functions, namely, imaging, PDT and PCI.	[123] 2021
Dendritic arginine-rich peptide conjugated with cystamine-modified stearic acid (ALS)	HPPH	Vascular endothelial growth factor (VEGF) siVEGF	HeLa	In vitro, Animals	Alternating low and high doses of light drove siRNA internalization and PDT, respectively. The liver and lung metastasis of HeLa cells were successfully inhibited, and the therapeutic effect was superior.	[58] 2021

**Table 3 pharmaceutics-13-01951-t003:** Codelivery of photosensitizers and immunotherapy drugs based on nanocarriers.

Nanoparticle	Photosensitizers	Immunotherapy Drugs	Tumor	Data Sources	Findings	Ref
Ce6-Gold nanoclusters-PEG2000-CD3 antibody-cytokine-induced killer cell NPs	Chlorin e6 (Ce6)	CD3 antibody	MGC-803	In vitro, Animals	The fluorescence intensity of GNCS-CE6 was approximately 4.5 times that of GNCs, thereby mediating a stronger PDT effect. Increased drug aggregation at the tumor site by targeting CIK cells.	[142] 2018
Human-induced pluripotent stem cells loaded with MnO_2_@Chlorin e6 NPs (iPS-MnO_2_@Ce6)	Chlorin e6 (Ce6)	Tumor antigens of human-induced pluripotent stem cells	Lewis	In vitro, Animals	iPS promoted the aggregation of nanoparticles at the tumor site and could be lysed by ROS produced by PDT to release tumor antigens. MnO_2_ interacted with H_2_O_2_ to release oxygen and alleviated tumor hypoxia.	[143] 2020
Chlorin e6- and imiquimod-loaded upconversion NPs (UCNPs-Ce6-R837 NPs)	Chlorin e6 (Ce6)	Imiquimod (R837)	CT26	In vitro, Animals	Lanthanide ions in UCNP nanoparticles could transform long near-infrared light into shorter wave light, improving tissue permeability. Toll-like receptor 7 agonist R837 was used as an adjuvant for enhanced antitumor immune response.	[144] 2017
Diblock copolymer azide-modified polyethylene glycol block polyaspartic acid(benzylamine) (Azide PEG-Pasp(Bz) micelles	Zinc phthalocyanine (ZnPc)	Mal-GGPLGVRG-Pra peptide modified aPD-L1	B16-F10	In vitro, Animals	Dual corresponding functions of pH and MMP-2 responses enhanced the aggregation of NPs in tumor cells. The blood circulation of NPs was prolonged, and their antitumor ability was enhanced.	[145] 2021
Serum albumin-coated boehmite B NPs	Chlorin e6 (Ce6)	Bee Venom Melittin (MLT)	4T1	In vitro, Animals	Significantly reduced the hemolysis caused by MLT and decreased the toxic side effects. MLT enhanced immunogenic cell death and dendritic cell activation, coactivating antitumor immunity with PDT.	[5] 2019

**Table 4 pharmaceutics-13-01951-t004:** Codelivery of photosensitizers and photothermal agents based on nanocarriers.

Nanoparticle	Photosensitizers	Photothermal Agents	Tumor	Data Sources	Findings	Ref
Boron dipyrromethene conjugated hyaluronic acid polymer NPs (BODIPY-HA NPs)	Boron dipyrromethene (BODIPY)	Boron dipyrromethene (BODIPY)	4T1	In vitro	BODIPY showed only PTT activity due to P-P stacking after self-assembly to form NPs. BODIPY-HA NPs were decomposed into BODIPY-HA molecules, restoring PDT activity and producing ROS after internalization by tumor cells.	[154] 2021
1,2-Distearoyl-sn-glycero-3-phosphoethanolamine-N-[carboxy(polyethylene glycol)-2000]-coated nanographene oxide-copper sulfide NPs (pGO-CuS NPs)	GO, CuS	Indocyanine green (ICG), CuS	MCF-7	In vitro	Photosensitizer-assembled NPs improved the drug loading efficiency and stability.	[155] 2015
Polypyrrole NPs	Phycocyanin (Pc)	Phycocyanin (Pc)	MDA-MB-231	In vitro	Pc produced PDT and PTT activity under 600 nm and 900 nm laser activation, respectively. It had good synergistic antitumor effects and cell imaging ability of PDT and PTT.	[156] 2017
Indocyanine green-coated single-walled carbon nanohorn NPs (SWNH-ICGs)	Indocyanine green (ICG)	Indocyanine green (ICG)	4T1	In vitro, Animals	The stability of ICG was improved, and ICG was protected from photodegradation. Under 808 nm laser irradiation, local hyperthermia and a large number of ROS were produced simultaneously, effectively inhibiting the growth of 4T1 tumors.	[157] 2018
Iridium oxide-manganese dioxide mineralized Chlorin e6 conjugated bovine serum albumin NPs (BSA-Ce6@IrO_2_/MnO_2_)	Chlorin e6 (Ce6)	Iridium oxide (IrO_2_)	MDA-MB-231, 4T1, PC3	In vitro, Animals	IrO_2_ and MnO_2_ decomposed endogenous H_2_O_2_ to alleviate tumor hypoxia and improve PDT. IrO_2_ presented excellent photothermal conversion efficiency (65.3%) and high X-ray absorption coefficient, enabling NPs to be used in computer CT and PA imaging.	[158] 2020
Lipid-purpurin 18 and pure lipid self-assembled NPs (Pp18-lipos)	Lipid-purpurin 18 (Pp18-lipids)	Lipid-purpurin 18 (Pp18-lipids)	4T1	In vitro, Animals	The Pp18-lipos with 2 mol% Pp18-lipids can perform PDT while with 65mol% can perform potent PTT. PTT/PDT synergistically inhibited tumor growth and enhanced tumor T cell immune response.	[159] 2020
Folic acid-polyethylene glycol-coated black phosphorus nanosheets conjugated with copper sulfide (BP-CuS-FA)	Black phosphorus (BP)	Copper sulfide (CuS), Black phosphorus (BP)	4T1	In vitro, Animals	FA could target tumor cells with FA receptor overexpression and enhance drug aggregation at tumor site. BP nanosheets could be degraded by ROS through oxidation processes to reduce toxic and side effects. BP-CuS-FA simultaneously mediated synergically enhanced the PDT-PTT antitumor effect and photoacoustic imaging.	[160] 2020
Poly(lactic-co-glycolic acid) (PLGA) NPs	IR780 iodide	Perfluorocarbon (PFC)	4T1	In vitro, Animals	PFC could be used as an artificial blood substitute to effectively increase hypoxia in the tumor microenvironment and enhance PTT and PDT. IR780 could simultaneously act as a PS and mediate mitochondrial targeting, disrupt the balance of mitochondrial ROS and induce irreversible apoptosis of tumor cells.	[161] 2020
Amino-modified nanomaterial based on MoS_2_ quantum-dot-doped disulfide-based SiO_2_ NPs coated with hyaluronic acid and chlorin e6 (MoS_2_@ss- SiO_2_-Ce6/HA)	Chlorin e6 (Ce6)	MoS_2_ quantum dots	4T1	In vitro, Animals	Effectively prolonged the blood circulation time of MoS_2_ quantum dots and increased the uptake of tumor cells. It could also be used for fluorescence/CT/MSOT image-guided PDT and PTT combination therapy.	[162] 2019
Isoindigo/triphenylamine donor-acceptor-donor conjugated small molecule NPs (IID-ThTPA NPs)	IID-ThTPA	IID-ThTPA	4T1	In vitro, Animals	The photothermal conversion efficiency reached 35.4%, and the singlet oxygen yield was 84%. The ultrahigh singlet oxygen quantum yield of IID-ThTPA NPs originated from the narrow singlet–triplet energy gap of IID-ThTPA.	[163] 2020
MoSe_2_/Bi_2_Se_3_ nanoheterostructure	MoSe_2_/Bi_2_Se_3_	MoSe_2_/Bi_2_Se_3_	HepG2	In vitro, Animals	ROS generation was promoted through photoinduced effective separation of electron-hole pairs. The photothermal conversion efficiency was increased to 59.3% by the nanoheterostructure. Displayed acid/photothermal sensitive drug release behavior.	[164] 2019
Polyethylene glycolated triphenylphosphine modified hitosan/iron oxide NPs (PEG-CS/Fe_2_O_3_ NPs)	Methylene blue (MB)	Iron oxide	HeLa, A549, MCF-7	In vitro, Animals	Improved the aggregation in the tumor site, reduce the toxic and side effects on normal tissues. Under low-power near-infrared light, NPs produced singlet oxygen, which can damage tumor cells.	[165] 2020
Indocyanine green-grafted gold nanobipyramids covalently conjugated with folic acid (AuBPs@FLA@ICG@FA NPs)	Indocyanine green (ICG)	Gold nanobipyramids (AuBPs)	B16-F10	In vitro	Enhanced the overall PTT-PDT efficiency by increasing the temperature by 2 °C and doubling ^1^O_2_ species generation.	[166] 2020
Pardaxin peptide-modified, indocyanine green-conjugated hollow gold nanospheres (FAL-ICG-HAuNS)	Indocyanine green (ICG)	Gold nanospheres	CT26	In vitro, Animals	FAL modification imparted endoplasmic reticulum targeting capability to NPs NIR irradiation induced strong ER stress and calcium reticulin (CRT) exposure, facilitating ICD-mediated immunotherapy. The antigen presentation function of dendritic cells was enhanced, and CD8+T cell proliferation and secretion of cytotoxic factors were activated.	[139] 2019
Gd^3+^ and chlorin e6 loaded single-walled carbon nanohorns (Gd-Ce6@SWNHs)	Chlorin e6 (Ce6)	Gd^3+^	4T1	In vitro, Animals	NPs could migrate from targeted tumors to tumors-draining lymph nodes, continuously activating DCs, and ultimately eliminating metastases.	[167] 2020

**Table 5 pharmaceutics-13-01951-t005:** Codelivery of photosensitizer and hyperthermia agent drugs based on nanocarriers.

Nanoparticle	Photosensitizers	Hyperthermia Agents	Tumor	Data Sources	Findings	Ref
Cancer cell membrane-cloaked Ce6-loaded Janus magnetic mesoporous organosilica NPs (CM@M-MON@Ce6)	Chlorin e6 (Ce6)	Magnetic mesoporous silica nanoparticles (M-MSNs)	4T1	In vitro, Animals	It exhibited the function of REDOX/PH double stimulation to induce PS release and matrix degradation. Increased the ability of targeted tumor aggregation and prolonged blood circulation time. Enhanced anticancer activity, simultaneously triggering a series of immunogenic cell death, resulting in a synergistic tumor-specific immune response.	[174] 2019
Ultramagnetic photosensitive liposomes	Foscan	Iron oxide NPs	SKOV-3	In vitro, Animals	The combination of photodynamic and magnetothermal therapy triggered the synergistic action of apoptotic signaling pathways, leading to complete eradication of tumor cells.	[175] 2015
Magneto low-density nanoemulsion (MLDE)	Chlorin e6 (Ce6)	Iron oxide NPs	MCF-7	In vitro	Improved the stability and biocompatibility of drugs. Identified the low-density lipoprotein receptor on the surface of tumor cells and improved the targeting selectivity of drugs.	[176] 2018

**Table 6 pharmaceutics-13-01951-t006:** Codelivery of photosensitizers and scintillators/radionuclides based on nanocarriers.

Nanoparticle	Photosensitizers	Scintillators/Radionuclides	Tumor	Data Sources	Findings	Ref
Hollow mesoporous silica NPs (HMSNs)	Chlorin e6 (Ce6)	Zirconium-89 (^89^Zr)	4T1	In vitro, Animals	^89^Zr activated Ce6 in the codelivery system and inhibited tumor growth. It did not rely on external light source and instead used internal radiation to achieve deep tumor treatment.	[194] 2016
Dextran modified TiO_2_ NPs (D-TiO_2_ NPs)	TiO_2_	Gallium-68 (Ga-68)	4T1	In vitro, Animals	The Cerenkov productivity of GA-68 was 30 times that of 18F-fluorodeoxyglucose. Ga-68 is an effective radionuclide for PDT, which can enhance DNA damage of tumor cells by codelivery.	[195] 2018
Magetic NPs with ^89^Zr radiolabeling and porphyrin molecules surface modification (^89^Zr-MNPs/TCPP)	Meso-tetrakis(4-carboxyphenyl)porphyrin (TCPP)	Zirconium-89 (^89^Zr)	4T1	In vitro, Animals	Under an external magnetic field, the NPs were highly concentrated in the tumor. Multimodal imaging of fluorescence, Cerenkov luminescence and Cerenkov resonance energy transfer could be implemented to detect the treatment process.	[196] 2018
Titanocene-modified, transferrin-coated TiO_2_ NPs (TiO_2_-Tf-Tc)	TiO_2_	Radiolabeled 2′-deoxy-2′-(18F)fluoro-D-glucose (FDG)	HT1080	In vitro, Animals	Low radiation induced radionuclide Cerenkov luminescence and activation of oxygen-independent photosensitizer TiO_2_, overcoming the limitation of the tumor hypoxia environment. Selectively destroyed a large number of tumor cells and induced lymphocyte infiltration.	[192] 2015
Folic acid-poly(lactide-co-glycolide) polymeric nanoparticles-verteporfin (VP), (FA-PLGA-VP NPs)	Verteporfin (VP)	Verteporfin (VP)	HCT116	In vitro	The NPs effectively killed HCT116 cells in the presence 6 MeV X-ray radiation. The 6 MeV X-ray radiation from LINAC produced energetic secondary electrons and Cerenkov radiation in the samples, which in turn excited the VP molecules.	[197] 2018
^131^I-labeled zinc tetra(4-carboxyphenoxy) phthalocyaninate conjugated Cr^3+^-doped zinc gallate NPs (^131^I-ZGCs-ZnPcC4)	Zinc tetra(4-carboxyphenoxy) phthalocyaninate (ZnPcC4)	Cr^3+^-doped zinc gallate (ZnGa_2_O_4_:Cr^3+^)	4T1	In vitro, Animals	^131^I could not only achieve long-term activation of PDT through Cerenkov luminescence and ionizing radiation but also directly kill tumor cells.	[193] 2021
LiLuF_4_:Ce@SiO_2_@Ag_3_PO_4_@cisplatin prodrug (Pt(IV)) NPs (LAPNP NPs)	Ag_3_PO_4_	LiLuF_4_:Ce	HeLa	In vitro, Animals	Pt(IV) acted as a sacrificial electrical receptor to enhance the yield of hydroxyl radicals by increasing the separation of electrons and holes in the photosensitizer. Pt(IV) generated cisplatin after receiving electrons, further enhancing the damage to tumor cells.	[198] 2018
NaGdF_4_:Tb,Ce@NaGdF_4_ core/shell structure NPs	Rose Bengal (RB)	NaGdF_4_:Tb,Ce	PC3	In vitro, Animals	The combination of X-ray guided photodynamic therapy and anaerobic oncolytic bacteria killed hypoxic and aerobic tumor tissue.	[199] 2020

**Table 7 pharmaceutics-13-01951-t007:** Codelivery of photosensitizers and sonophotosensitizers based on nanocarriers.

Nanoparticle	Photosensitizers	Sonophotosensitizers	Tumor	Data Sources	Findings	Ref
Glypican-3-targeted, curcumin-loaded microbubbles (GPC3-CUR-MBs)	Curcumin (CUR)	Curcumin (CUR)	HepG2	In vitro, Animals	The combined therapy sonophotodynamic therapy (SPDT) had more obvious antitumor effects than SDT or PDT alone.	[213] 2020
Curcumin-loaded poly(L-lactic-co-glycolic acid) microbubbles (CUR-PLGA-MBs)	Curcumin (CUR)	Curcumin (CUR)	HepG2	In vitro	Mitochondrial membrane potential loss was induced by increased reactive oxygen species (ROS) generation, leading to tumor cell apoptosis and thermal decay.	[214] 2020
5-Aminolevulinic acid/titanium dioxide NPs (5-ALA/TiO_2_)	5-Aminolevulinic acid (5-ALA)	Titanium dioxide (TiO_2_)	SCC	In vitro, Animals	Using countercurrent chromatography to obtain regular size nanoparticles enhanced the efficacy of codelivery of PDT and SDT for tumor treatment.	[215] 2016

**Table 8 pharmaceutics-13-01951-t008:** Codelivery of multidrugs based on nanocarriers.

Nanoparticle	Type of Combination Therapy	Tumor	Data Sources	Findings	Ref
Human serum albumin-paclitaxel-sinoporphyrin sodium nanotheranostics (HAS-PTX-DVDMS)	Photodynamic therapy/Chemotherapy/Sonodynamic therapy	4T1	In vitro, Animals	As a cosolvent, HAS improved the water solubility of the photosensitizer and dispersed it sufficiently to reduce the aggregation quenching effect. It could be used as fluorescent probe, with stronger fluorescence imaging ability and ^1^O_2_ generation ability than that of DVDMS alone. The addition of ultrasound increased the tumor cell mortality rate from 70% to 90%, and the effect of triple therapy was superior.	[222] 2020
ZrO_2_-coated, doxorubicin hydrochloride-, chlorin e6- and tetradecanol-loaded upconversion NPs (UCNPs@ZrO_2_-Ce6/DOX/PCM)	Photodynamic therapy/Chemotherapy/Hyperthermia	U14	In vitro, Animals	It could be used to realize multimodal imaging guidance (MRI, CT and UCL) in conjunction with tumor therapy. Under near-infrared light, the heat generated by UCNPs helped to kill cancer cells and dissolve PCM, triggering the release of drugs and the production of ROS, resulting in an effective integration of photothermal/photodynamic therapy with chemotherapy.	[223] 2017
Iron-dependent artesunate-loaded, transferrin-modified, hollow mesoporous CuS NPs (AS/Tf-HMCuS NPs)	Photodynamic therapy/Photothermal therapy/Chemotherapy	MCF-7	In vitro, Animals	TF-mediated endocytosis enhanced accumulation and retention in tumor cells. Effective near-infrared light was converted to heat while producing high levels of ROS for PDT.	[224] 2017
1-Tetradecanol-, doxorubicin- and chlorin e6-loaded hollow mesoporous copper sulfide NPs (H-CuS@PCM/DOX/Ce6 (HPDC) NPs)	Photodynamic therapy/Photothermal therapy/Chemotherapy	4T1	In vitro, Animals	Under near-infrared laser radiation, the NPs produced a strong photothermal effect, which induced controlled release of DOX and Ce6. The NPs presented low systemic toxicity and good blood compatibility and were able to eradicate breast cancer in 4T1 mice.	[225] 2019
(Enaminitrile molecule gel encapsulating doxorubicin core/Mesoporous silica-coated, CuS-loaded, lanthanide ion-doped upconversion shell) NPs wrapped with a cancer cell membrane	Photodynamic therapy/Photothermal therapy/Chemotherapy	MCF-7, 4T1	In vitro, Animals	Wrapping NPs in cancer cell membranes imparted unique advantages such as immune escape and homologous binding ability. Through the transformation of near-infrared energy into ultraviolet light from the UCNP nucleus, enaminitrile molecule phase transition was stimulated to generate ROS for PDT, thus avoiding the limited penetration of ultraviolet light to tissues.	[226] 2021
(Upconversion core/chlorin e6, doxorubicin hydrochloride coloaded mesoporous silica shell) NPs conjugated with polyethylene glycol-modified graphene (DOX-UMCG)	Photodynamic therapy/Photothermal therapy/Chemotherapy	HeLa	In vitro, Animals	Only low-intensity near-infrared light was required to produce a domino effect, avoiding the phototoxicity of high-intensity exposure to nearby healthy cells. Synergistic Chemo/PTT/PDT anticancer effect with low systemic toxicity.	[227] 2020
Thiol-terminated monomethoxyl poly(ethylene glycol) and mercaptoropionylhydrazide-modified gold nanorods covalently conjugated with 5-aminolevulinic acid and doxorubicin (GNRs-MPH-^ALA/DOX^-PEG)	Photodynamic therapy/Photothermal therapy/Chemotherapy	MCF-7	In vitro, Animals	Combined CT/PDT/PTT therapy killed McF-7 cells more effectively, with superadditive antitumor effects and no obvious systemic toxicity. The circulating half-life of GNRs-MPH-^ALA/DOX^-PEG in blood was approximately 52 min, and the tumor aggregation rate was 3.3%.	[228] 2018
Folic acid-functionalized, paclitaxel-loaded MgAl layered double hydroxide gated mesoporous silica NPs (MT@L-PTX@FA)	Photodynamic therapy/Photothermal therapy/Chemotherapy	HepG2	In vitro	Under near-infrared irradiation, the NPs effectively converted photon energy into ROS and heat, enhancing toxicity to tumor cells. It presented obvious slow-release characteristics and was sensitive to pH. It could dissolve MgAl LDHs into Mg^2+^ and Al^3+^ under the low pH of the tumor site, selectively releasing PTX for chemotherapy.	[229] 2019
Folic acid-CuS/docetaxel@polyethylenimine-protoporphyrin IX-CPG (FA-CuS/DTX@PEI-PpIXCpG)	Photodynamic therapy/Photothermal therapy/Chemotherapy/Immunotherapy	4T1	In vitro, Animals	FA-CD@PP-CpG medium and low-dose DTX promoted CTL infiltration, improved the efficacy of anti-PD-L1 antibody (APD-L1), inhibited MDSCs, and effectively polarized MDSCs to the M1 phenotype.	[220] 2019

## Data Availability

Not applicable.

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
