# Peer review of "Progress in Nanocarriers Codelivery System to Enhance the Anticancer Effect of Photodynamic Therapy"

_pharmaceutics, 2021, doi:10.3390/pharmaceutics13111951_

Round 1
Reviewer 1 Report
The Abstract lists potential limitations of PDT. It is not clear what is meant by ‘long term tolerance’. I doubt that reactive oxygen species are going to lose their ability to damage biologic systems. The introduction appears to indicate that a laser is necessary for the photodynamic process, but this is not correct. Any light source will be adequate. While some applications do involve lasers, many involve other light sources. While the list of applications is comprehensive, interpretations are often incorrect.
The description of photosensitizing agents is peculiar. The only requirement is that the system produces reactive oxygen (and nitrogen) species upon irradiation. While some photosensitizers are related to the porphyrin structure, others are not. This is not really a relevant consideration. The division of the group into first, second and third (etc) generation agents is arbitrary. It is true that the earliest work was done with ‘hematoporphyrin derivative’, a complex mixture of porphyrins. In a search for less complex systems, chemists began exploring the preparation of individual agents including porphyrins, reduced porphyrin systems, i.e., bacteriochlorins, phthalocyanines, and other structures. The goal was promoting efficacy by searching for agents with significant absorbance at longer wavelengths, and avoiding use of complex mixtures. It is, for example, incorrect to claim that ‘these compounds have a higher tissue penetration depth’ (line 46). It is the activating light that has the greater tissue penetration depth. Low water solubility does not limit IV administration. Formulation techniques can readily circumvent this process, e.g., the procedure whereby BPD (benzoporphyrin derivative, Visudyne) is prepared for administration. I also do not understand what ‘rapid aggregation’ means (lines 50-51).
I assume that Fig. 12 refers to the use of PDT in assorted combinations. The idea of damaging tumor cell DNA (panel 1) likely refers to the initiation of apoptosis. Otherwise, damaging DNA can lead to mutations which is not a good idea.
Section 2.1 reveals many basic misunderstandings. For example, there appears to be a distinction between ROS and singlet oxygen. Singlet oxygen IS a reactive oxygen species. Ascribing a role to the tissue dielectric constant: what does this mean? Apoptosis does not result in necrosis. It results in cell fragmentation and AVOIDS necrosis. Autophagy is most often a cytoprotective process. In 2.2.1, it is claimed that HPD cannot be used for ‘deep tumor tissue’, but with the aid of fiber optics and lasers, this is readily feasible. Curcumin is useless for PDT except in cell culture since the activating wavelength (430 nm) will not penetrate tissues beyond one or two cell diameters.
It is not clear where the idea that phthalocyanines cannot target tumors (2.2.5) since there are many examples of successful treatment of animal models. The persistent photosensitization of skin was observed with HPD and Photofrin because of their complex nature. While there are few examples relating to the use of other agents in clinical practice, BPD does not appear to have this problem. Solubilization of BPD involved detergents, not necessarily nanocarriers.
The summary of assorted procedures for use in therapy looks to be quite comprehensive. The Outlook section has problems. The great limitation of PDT is the requirement for light. The problem is not with ‘deep’ tumors but with metastatic tumors where the site is unknown. A complete ablative effect HAS been noted and there are many examples of curative treatment where the tumor has not metastasized. The references relating to combinations of PDT + chemotherapy and immunology neglect the significant contribution of Hasan’s group to this topic. Examples: Photochem Photobiol 95, 1288-1305, 2019; Nanoscale 7, 12471-12503, 2016; Cancers 12, 1401, 2020; J Clin Med 9, 2390, 2020; Nano Lett 19, 7573-7587, 2019.
Reviewer 2 Report
The review entitled “Codelivery against Cancer: Photodynamic therapy and what? by YuLing Yang, Ke Lin and Li Yang, reports recent development concerning the combinations of PDT with other cancer therapies like chemotherapy, gene therapy, immunotherapy, photothermal therapy, hyperthermia, radiotherapy, sonodynamic therapy and multidrug therapy. The topic is actual, has interest and is well organized. However, some reformulations and clarifications are required.
An important aspect is related with the difficulty to understand in some comments which is the photosensitiser since in most of the cases the authors just use abbreviations without referring the full meaning. Below some alert are given but the authors must revise the article in order to improve this aspect.
The considerations are presented according with the appearance of comments in the manuscript and not by their relevance:
- The title needs to be revised since does not reflect what it is discussed.
- Line 22: Instead “of codelivery nanocarriers” probably the authors want to say “using codelivery nanocarriers”
- The Keywords must be revised. Suggestions: Codelivery nanocarriers, photodynamic therapy, anticancer therapies, combining therapies.
- Line 36 add recent healing articles to references concerning the use of PDT (e.g. https://doi.org/10.3390/cancers13205176; IntechOpen: London, UK, 2017; Volume 5, pp. 75–94; ISBN 978-953-51-3156-4. 54; J. Mol. Sci. 2020, 22, 234; https://doi.org/10.3390/ijms22084121 among others).
- Line 37 Concerning the following comment improve the English since saying that porphyrin type PS has been developed for three generations is not correct. “The PS types are generally divided into porphyrins and nonporphyrins. The latter type is typical of hypericin and has lagged behind in development, while the former has been developed for three generations.”
- Line 43. Try to improve the meaning of the comment by consulting excellent reviews on the subject: “Hematoporphyrins, the first generation of PS and the first used in the clinic, were isolated from hemoglobin. Limitations such as excessive accumulation of light in the skin and a long half-life lead to prolonged exposure to light during treatment, which can mean major changes in patients’ lifestyle”
- Line 51 The word aggregation is not ideal probably it is better to use accumulation or uptake.
- Line 70 Figure 1 must come before the references. Check similar situations in other parts of the manuscript.
- In figure 1 check and correct in the central picture the reference to type I and type II mechanism.
- Line 52 Singlet oxygen is also a ROS so reformulate the phrase in order to show this.
- Line 107 add: and on the structural features of the PSs.
- Line 121 check the name of hematoporphyrin by consulting revisions for an adequate description of the PSs, (e.g. DOI: 1590/0001-3765201820170811) but of course there are many more.
- Line 127 instead of zone use band.
- Line 134. Change to as antibacterial [27] or antiviral [28] therapeutic agent among other applications.
- Line 148 Since there is more than one phthalocyanine is better to put this word in plural.
- From Line 151, clarify the meaning of the following comment: “The strength of phthalocyanine is two orders of magnitude higher than that of most porphyrins [38], and it has ….” And of the other comments in this topic since they are too entropic.
- Line 168 correct the structure of 1O2 and verify all the manuscript since the same mistake appear in other parts. Add recent articles concerning this topic namely from Pharmaceutics (e.g. 3390/pharmaceutics13091512 and others ).
- Line 256: Check the comment since seems strange “The introduction of porphyrin-phospholipid (PoP) coupled with PS enables its photoprogramm ..”
- Line 266 “while loading irinotecan” what it is it ?
- Line 280 what is IR780?
- From now on the manuscript lost the indication the lines so they can not be indicated: In the comment “Another type of photoprogrammed gene regulation uses PDT active nanomaterials as gene delivery vectors, such as PPBP, a black scale nanomaterial that shows PDT activity under light and then specifically degrades to release siRNA in a high ROS and acidic tumour environment to achieve targeted delivery [107].” What is PPBP?
- Clarify the comment: Deng S [110] used CRISPR–Cas9 technology to edit the antioxidation regulator Nrf2 and increase the sensitivity of tumor cells to ROS produced by PDT.
- Which was the PS in the comment concerned the work of “Zheng WH [109] used anisamide-targeted lipid-calcium-phosphate (LCP) nanoparticles to achieve codelivery of PS and HIF-1α siRNA. …”
- It is difficult to understand the material developed by reported in Cao Y [112],
- Cai Z [126] prepared PCN-ACF-CpG@HA NPs loaded with the immune adjuvant CPG by combining PS H2TCPP with zirconium ions to target tumors with high expression of the CD44 receptor. What is H2TCPP?
- Section 4.4. In the comment “Another more conventional codelivery method is to simultaneously load the photothermal agent and PS in nanodelivery systems such as micelles, vesicles, and liposomes. Gang Chen et al. [149] developed CS NPs as codelivery carriers of photothermal agents (IR780) and PS 5-aminolevulinic acid (5-ALA) for oral administration in the treatment of subcutaneous colon cancer in mice” be careful because 5-aminolevulinic acid (5-ALA) is not a PS but a prodrug that is metabolized in protoporphyrin IX
- In abbreviations ZnF16Pc is not zinc phthalocyanine. Correct the name.
Round 2
Reviewer 1 Report
The title of this report suggests that the intent is to discuss the use of nanocarriers for enhancing efficacy of photodynamic therapy in a ‘codelivery’ context.. The content often diverges substantially from the title. I suggest that the authors eliminate the extraneous material and concentrate on a discussion of the advantages of combining PDT with other modalities. Since there are many excellent reviews on the topic of PDT, it is not necessary to recapitulate all of the details. An extended discussion of PDT is unnecessary.
While the abstract and other material suggests that PDT alone is inadequate for cancer control, there are numerous examples indicating that this is not true. The limitations of PDT (poor tissue optics, metastatic disease, hypoxia) are well known. Choosing the optimal subcellular PDT targets can be a potentially significant advance but this is not considered here. One example is a report showing a marked improvement in PDT efficacy by expanding the range of sub-cellular targets, e.g., Photochem Photobiol. 95, 419-429 (2019). There are others.
The abstract (lines 16-17) mentions ‘curative effect decreases after long-term treatment’; the meaning is unclear. What ‘long term treatment’ is used for PDT? Protocols usually involve a single treatment: photosensitization and irradiation. Do the authors suggest that malignant cell types eventually become ‘resistant’ to photodamage? Figure 1 panel A: it is not clear that chemotherapy necessarily increases ROS production.
Many items discussed in this report are not explained. For example (line 86), PDT will have no effect on metastatic cancer if the site is unknown and therefore not irradiated. As noted above, the major issues contributing to the failure of PDT are metastatic tumor at unknown sites, inadequate light delivery and lack of sufficient oxygen. It is not necessarily true that singlet oxygen (line 132) is the ‘most significant’ of the ROS. Many very effective agents produce other oxygen radicals but not singlet oxygen. The role of the tissue dielectric constant (line 133) is unclear. What does this mean? Material contained in lines 135-145 is very confusing and irrelevant to the use of PDT combinations. I assume that the point is that the sub-cellular sites and localization are important. This is true: damage to lysosomes or mitochondria can induce apoptosis. Damage to the ER can result in apoptosis, paraptosis or both. Since lysosomes are involved in the cytoprotective process termed autophagy, inhibiting autophagy this can also promote photokilling. This is not well explained.
The material in section 2.2 is somewhat limited and ignores many of the most successful PDT agents like benzoporphyrin derivative (BPD, Visudyne). At this point, a very large number of photosensitizers have been identified. Only a few have been successfully used in clinical practice. Why curcumin should be included is unknown since the 430 nm absorbance optimum will make this useless for PDT except in cell culture. What does ‘at the level of tumor cells’ mean (line 180)? A discussion of select agents is not really necessary unless this can somehow be related to the topic: use of combined modalities to promote PDT efficacy.
The ‘EPR’ effect is mentioned although it is not entirely clear how is affected by nanocarriers. It is true that directing photosensitizers to specific sub-cellular loci can promote efficacy, as discussed by some recent reports by Obaid et al, but these are not discussed.
It is indicated that PCI is a form of gene therapy (section 4.2). Line numbering is lost at this point. None of this is consistent with PCI where a photodynamic effect permits the distribution of chemotherapeutic agents to malignant tumor sites. This does not involve genes. The tables do not distinguish among in vitro results, studies involving animals and clinical data. It is relatively easy to kill tumor cells in culture. While such results can point the way to in vivo work, readers will wonder whether experimental data are confined to work in cell culture or whether there appears to be any indication that combined modalities and effects of nano-delivery results in improved vivo efficacy.
Other items: check spelling of apoptosis (Fig. 3); Cerenkov (often misspelled) throughout; Fig. 1 panel F ‘killed’;
Reviewer 2 Report
In the revision of the review by YuLing Yang, Ke Lin and Li Yang, the authors had in account in general the comments but there is still need to correct some considerations
Line 53 remove the word and before etc.
Line 54 correct to inorganic nanoparticles
Line 56 Porphyrins PSs in the early stage were mainly mixtures obtained from natural porphyrins, such as porphyrins derivatives, etc What the authors want to say about porphyrins derivatives. Probably is better to eliminate in natural porphyrins.
Line 60 Verify the comment porphyrin chlorobenzene and porphyrin phthalocyanine. This is not correct.
Line 138 This comment seems not finished. “which is more effient than mitochondrial”
Figure 1 is not in the correct place.
Line 163 Use Chlorin in plural Chlorins
Line 166 check the english
Line x? check in the new comment “monoclonal antibody Cetuximab with biosynthesis of benzoporphyrin derivative for pancreatic ductal adenocarcinoma treatment.” The word biosynthesis of benzoporphyrin.
Round 3
Reviewer 1 Report
This is much improved and appears to be a comprehensive review of the topic indicated.
At this point, there are only minor issues. In the ‘Hyperthermia Therapy’ section, how does hyperthermia increase cellular uptake of oxygen molecules?
There are a few spelling items to be checked, e.g., line 56 ‘concentrationand’ needs to be changed to ‘concentration and’.
There is no mention of the ability of the fluorescence of photosensitizers to aid in identifying tumor loci. While this is not a topic related to the PDT efficacy, it might be mentioned.
The only other topic of concern is over-use of the term ‘laser’, e.g., in line 358. This gives an impression that that there is something special about laser light. Since this is monochromatic, and can be produced at a significant intensity and is easily directed, lasers are often the preferred method for delivering light. But there is nothing special about laser-generated photons. Any light source will do. Thus, line 355 could be replaced with’ROS produced by irradiation’ with the word ‘laser’ eliminated. This is also true elsewhere, e.g., the legend to Fig. 4 where ‘laser’ can be replaced with ‘light’.
While not a matter of great concern, the headings of the tables tend to contain the words ‘More examples of recent studies on . . .’ This phrase can be eliminated. Readers will realize that what is being reported are examples.
Reviewer 2 Report
The authors improve the manuscript that it is now more concentrated on the advantages of combining PDT with other treatment modalities
